# Compressed ultrahigh-speed single-pixel imaging by swept aggregate patterns

Patrick Kilcullen [1], Tsuneyuki Ozaki[1] & Jinyang Liang [1] ✉

Single-pixel imaging (SPI) has emerged as a powerful technique that uses coded wide-field illumination with sampling by a single-point detector. Most SPI systems are limited by the refresh rates of digital micromirror devices (DMDs) and time-consuming iterations in compressed-sensing (CS)-based reconstruction. Recent efforts in overcoming the speed limit in SPI, such as the use of fast-moving mechanical masks, suffer from low reconfigurability and/or reduced accuracy. To address these challenges, we develop SPI accelerated via swept aggregate patterns (SPI-ASAP) that combines a DMD with laser scanning hardware to achieve pattern projection rates of up to 14.1 MHz and tunable frame sizes of up to 101×103 pixels. Meanwhile, leveraging the structural properties of S-cyclic matrices, a lightweight CS reconstruction algorithm, fully compatible with parallel computing, is developed for real-time video streaming at 100 frames per second (fps). SPI-ASAP allows reconfigurable imaging in both transmission and reflection modes, dynamic imaging under strong ambient light, and offline ultrahigh-speed imaging at speeds of up to 12,000 fps.

Single-pixel imaging (SPI) is a potent computational imaging modality with widespread applications[1]. This approach imparts deterministic two-dimensional (2D) spatial patterns onto the imaging beam, followed by data acquisition using a single-pixel detector. In the ensuing image reconstruction, these spatial patterns are used as prior knowledge to facilitate the accurate recovery of spatial information. By eliminating the need for a 2D sensor (e.g., CCD or CMOS), SPI may use detectors whose cutting-edge performance or high specialization are impractical to manufacture in an array format[2]. For example, SPI paired with photomultiplier tubes has been used for time-of-flight three-dimensional (3D) profilometry[3,4], single-photon imaging[5], and imaging of objects hidden from direct line-of-sight[6]. SPI combined with electro-optic detection has been used in terahertz (THz) imaging[7–11]. SPI implemented with ultrasonic transducers has found applications in photoacoustic imaging[12,13].

Recent years have witnessed the growing popularity of SPI owing to the hardware progress in spatial light modulators (SLMs) and software advances in image reconstruction. To date, the dominant SLM used in SPI is the digital micromirror device (DMD), which is capable of reconfigurable and nonmechanical display of mega-pixel patterns at refresh rates of up to 32 kHz[1]. However, despite offering numerous advantages, their kHz pattern refresh rates sharply limit the data acquisition rate of most SPI systems below the MHz bandwidths of typical photodetectors. Consequently, the 2D imaging speeds in most SPI systems are clamped under 100 frames per second (fps), precluding them from high-speed imaging.

To compensate for DMDs' limitation in pattern projection rates, the majority of SPI approaches have aimed to increase imaging speeds by reducing the number of samples required for image recovery using compressed sensing (CS)[14–17]. While widely adopted in SPI, typical CS algorithms are implemented in the framework of iterative convex optimization, which requires a large computational overhead with non-deterministic processing times highly dependent on the scene complexity[18–20]. While it is suitable for high-speed imaging experiments with offline processing, the excessively long image reconstruction time (typically tens of seconds) poses inherent challenges for real-time visualization, where reconstruction must immediately follow acquisition.

[1]Centre Énergie Matériaux Télécommunications, Institut National de la Recherche Scientifique, Université du Québec, 1650 boulevard Lionel-Boulet, Varennes, QC J3X 1P7, Canada. ✉e-mail: jinyang.liang@inrs.ca

Towards this goal, many ensuing developments have substituted DMDs with new light sources and modulators[21–24]. One recent approach has used a high-speed illumination module comprised of an array of 32×32 light-emitting diodes (LEDs)[23,24]. Although this method has demonstrated a pattern deployment rate of 3.13 MHz and a CS-assisted 2D imaging rate of 25 thousand frames per second (kfps), the performance of its patterned light source, which requires custom hardware, is closely tied to the display of Hadamard patterns that possess certain symmetry properties. Moreover, because the patterns were directly provided by LED illumination, this approach is inapplicable to many types of SPI requiring either passive detection[25] or specialized light sources, such as lasers[26].

Alternatively, the speed of SPI systems can be enhanced by using patterned transmissive masks mechanically scanned at high speeds[22,27]. Coded patterns etched onto printed circuit boards have been used in THz and millimeter-wave SPI for over a decade[28–33]. Systems of this sort frequently employ fast rotating disks to impart spatial modulation to the illumination[34,35]. To enhance pattern deployment from continuous rotation, much emphasis in pattern design and reconstruction has been placed on the use of cyclic matrices[36–40], which allow for the deployment of a new masking pattern from a translational shift of only one pattern pixel[41]. The latest development in this approach reported a pattern projection rate of 2.4 MHz, which transferred to a non-CS recovery of a frame size of 101×103 pixels at 72 fps[22]. In another study[27], assisted by CS and optimization-based image reconstruction, the imaging speed reached 100 fps despite a reduced frame size of 32×32 pixels. However, in contrast to the advantages of reconfigurable DMDs, these physical masks often increase the systems' complexity and considerably reduce their flexibility. For example, the inherent trade-off between pixel count and pixel size for fixed disk dimensions suggests an increased difficulty in manufacturing for larger encoding masks. Consequently, auxiliary tools, such as a synchronized steering mirror[22], may be required to correct inaccurate radial motion caused by the non-concentricity of the patterns.

To overcome these limitations, we develop single-pixel imaging accelerated via swept aggregate patterns (SPI-ASAP). The implementation of laser scanning rapidly deploys individual encoding masks as optically selected sub-regions of larger aggregate patterns that are displayed via DMD. In this way, pattern aggregation enhances SPI data acquisition rates to more than two orders of magnitude above the limitations of DMD-only modulation, while still retaining the high pixel count and flexibility of pattern display with DMDs. Meanwhile, a fast CS reconstruction algorithm, which is tightly coupled with the architecture of pattern deployment, is developed with full compatibility with parallel computing for real-time visualization. Altogether, SPI-ASAP projects encoding patterns at a rate of up to 14.1 MHz, allowing for reconfigurable 2D imaging offline at up to 12 kfps. Unrestricted by the iterative optimization of conventional CS-based image reconstruction, SPI-ASAP empowers real-time video operation at 100 fps for frame sizes of up to 101×103 pixels.

## Results
### System setup

A schematic of the SPI-ASAP system is shown in Fig. 1. A 200 mW continuous-wave laser operating at 671 nm is used as the light source. The ultrahigh-speed pattern sweeping is generated by a 16-facet polygonal mirror, a 0.45" DMD, and associated optical components (lenses L1-L5 in Fig. 1). The incident beam is first reflected by a facet of the polygonal mirror, which creates illumination that rapidly moves across the DMD surface at a 24° angle of incidence. Binary patterns, represented numerically as {0,1}-valued 2D arrays, are displayed on this DMD. Containing an array of 912×1140 square micromirrors (7.6 μm pitch), the DMD uses tilt actuation to either discard the beam (via the 0-valued pixels) or reflect it (via the 1-valued pixels) back to the polygonal mirror. An iris, positioned near the back focal plane of L4, selects the strongest diffraction order. Owing to the symmetry of the optical path, the second reflection by the polygonal mirror imparts an equal and opposite scanning motion to the imaging beam, which stabilizes its position at an adjustable slit. Meanwhile, because this slit receives an image of the illuminated part of the DMD's surface, stabilization by the second reflection also imparts a rapid scanning motion to the patterns that tracks with the motion of the scanned illumination. This dual-scanning operation increases the optical efficiency of SPI-ASAP over that of a single-scanned design because illumination is concentrated on only the parts of the DMD surface that are conveyed to the adjustable slit.

During operation, each scan induced by a polygonal mirror facet is synchronized with the display of a new aggregate binary pattern pre-stored by the DMD. After transmitting through the slit, the structured illumination interrogates the structure of objects placed in the image

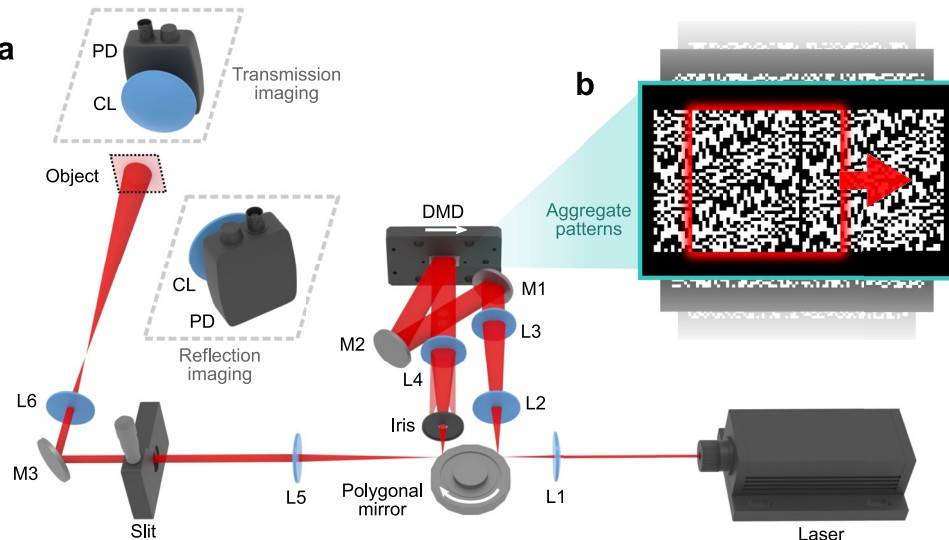

**Fig. 1 | Schematic of SPI-ASAP. a** System illustration showing beam scanning of a DMD-displayed aggregate pattern by rotation of a polygonal mirror. White arrows indicate directions of polygonal mirror rotation and illumination scanning. CL, condenser lens; DMD, digital micromirror device; L1-L6, lenses; M1-M3, mirrors; PD, photodiode. **b** Closeup of an example aggregate pattern in a sequence suitable for imaging at a frame size of 41×43 pixels. Red box: a representative encoding pattern. Red arrow: scanning direction.

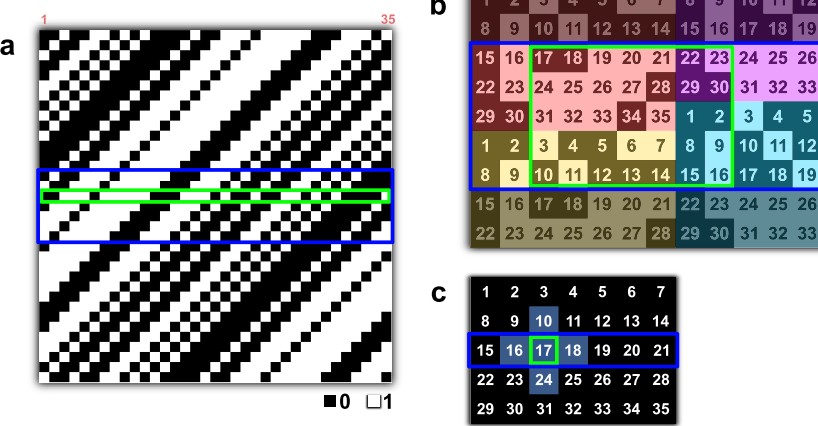

**Fig. 2 | Coding strategy of SPI-ASAP. a** Binary cyclic S-matrix of order $n = 35$ (i.e., $p=7$ and $q=5$). Green boxes (also appearing in (**b**) and (**c**)) highlight the role of a single encoding pattern. Purple boxes (also appearing in (**b**) and (**c**)) highlight a range of $p$ rows deployed together in each scan. **b** Restructured matrix pattern formed from the matrix shown in (**a**). The numbers represent the corresponding column indices of each element of the first row of the matrix shown in (**a**), with shaded colors highlighting four tiled copies of the reshaped initial row. The purple-boxed region shows an aggregate pattern that is displayed via DMD. The green-boxed region shows the single encoding pattern (corresponding to row 17) also highlighted in (**a**). **c** Arrangement of the bucket signals as a 2D array with numbers corresponding to row indices of the matrix in (**a**). Gray shading highlights an expected neighbourhood of similar values surrounding the measurement from the encoding pattern highlighted in green in (**b**). The purple-boxed region shows a row of bucket signals acquired from the deployment of the aggregate pattern highlighted in (**b**).

plane (imaged by lens L6). A portion of light transmitted or reflected by an object is focused by a condenser lens to a photodiode that is positioned to receive light in either the transmission mode or the reflection mode. The recorded data points, referred to as "bucket signals", are transferred to a computer for ensuing image processing. The full details of the experimental setup, principles of laser beam encoding and de-scanning, as well as system synchronization, are discussed in Methods, Supplementary Notes 1–2, and Supplementary Figs. 1–2. A discussion of the advantages of polygonal mirror scanning in SPI-ASAP, as well as a performance comparison with previous scanning SPI systems, is provided in Supplementary Note 3, Supplementary Fig. 3, and Supplementary Table 1.

## Coding strategy and image recovery

The bucket signals, denoted by an $m$-element vector **y**, can be regarded as the optically computed inner products between an $n$-pixel image **x** and the set of encoding patterns $\{\boldsymbol{s}_i, i = 1, \ldots, m\}$. This process can be expressed in the form of a single $m$ by $n$ matrix equation

$$\mathbf{y} = S\,\mathbf{x}, \tag{1}$$

where the measurement matrix $S$ contains each encoding pattern written in row form and corresponding to the order of the bucket signals in **y**. SPI-ASAP uses cyclic S-matrices to generate encoding patterns, with Fig. 2a illustrating an example for $n = 35$. Individual encoding patterns arise from the 2D reshaping of the rows of $S$ by row-major ordering. Writing $n = pq$ for the dimensions of the reshaped rows, it is possible to restructure the information in $S$ to form a $(2p - 1) \times (2q - 1)$ pattern that exactly encompasses all 2D encoding patterns determined by its rows. An example of such restructuring is shown in Fig. 2b, from which it can be seen that each $p \times q$ sub-region represents a 2D reshaped row of $S$, with the corresponding row index appearing as the number in the top leftmost corner element. In each scan, a $p \times (2q - 1)$ sub-region of the restructured matrix pattern displays on the DMD. The scanning action of the optical system then sequentially illuminates and projects each $p \times q$ sub-area of this pattern, resulting in the deployment of $q$ encoding patterns (see an example in Fig. 2b). This process then repeats, each time using a different $p \times (2q - 1)$ sub-region of the restructured matrix pattern to

aggregate the deployment of $q$ encoding patterns in each scan. After $p$ scans, a full measurement is completed when the DMD-displayed aggregate patterns traverse through the entire restructured matrix pattern. More details of cyclic S-matrices are discussed in Methods.

Owning to the 2D packing relationship of the encoding patterns within the restructured matrix pattern, their deployment in SPI-ASAP can be regarded as an operation of 2D discrete convolution on the $(2p - 1) \times (2q - 1)$ restructured matrix pattern, with the underlying image used as a kernel of size $p \times q$. Thus, because of the high similarity in the structures of spatially adjacent encoding patterns, the bucket signals **y** exhibit 2D smoothness when reshaped to a matrix $Y$ of the same size as the image (Fig. 2c). Moreover, periodic boundary conditions are exhibited by $Y$, allowing 2D smoothness to extend to all elements. Therefore, in general, a bucket signal indexed to row $i$ of $S$ will exhibit a high similarity with the bucket signals that surround it in $Y$, with the four nearest neighbouring bucket signals in particular corresponding to the rows $i - 1$, $i + 1$, $i - q$, and $i + q$ in $S$, with $q$ denoting the width of the reshaped rows and indices interpreted modulo $n$.

This property motivates a strategy to incorporate CS (i.e., $m < n$) in rapid image recovery. By using carefully selected encoding patterns, a set of evenly distributed samples is selected on $Y$. Then, the values of non-sampled elements can be estimated by interpolation, which empowers image recovery by direct inversion of Eq. (1). The architecture of SPI-ASAP enables a particular sampling strategy in which bucket signals corresponding to complete rows of $Y$ are acquired in each scan. Aggregate patterns are thus formed in the manner shown by the purple boxes in Fig. 2, with the displayed DMD pattern sequence determined by an evenly distributed selection of rows from $Y$.

In this case of row-wise subsampling, non-sampled data in $Y$ can be approximated by one-dimensional (1D) interpolation along columns with a method such as spline interpolation. This operation, along with optional low-pass filtering, can be implemented using only matrix multiplication with the assistance of pre-computed elements. Thus, SPI-ASAP's reconstruction algorithm is compatible with the architecture of a graphic processing unit (GPU) for real-time visualization. Meanwhile, by sectioning streams of bucket signals generated by continuous scanning, videos can be reconstructed with tunable frame rates. Details regarding the sequencing of aggregate patterns, data

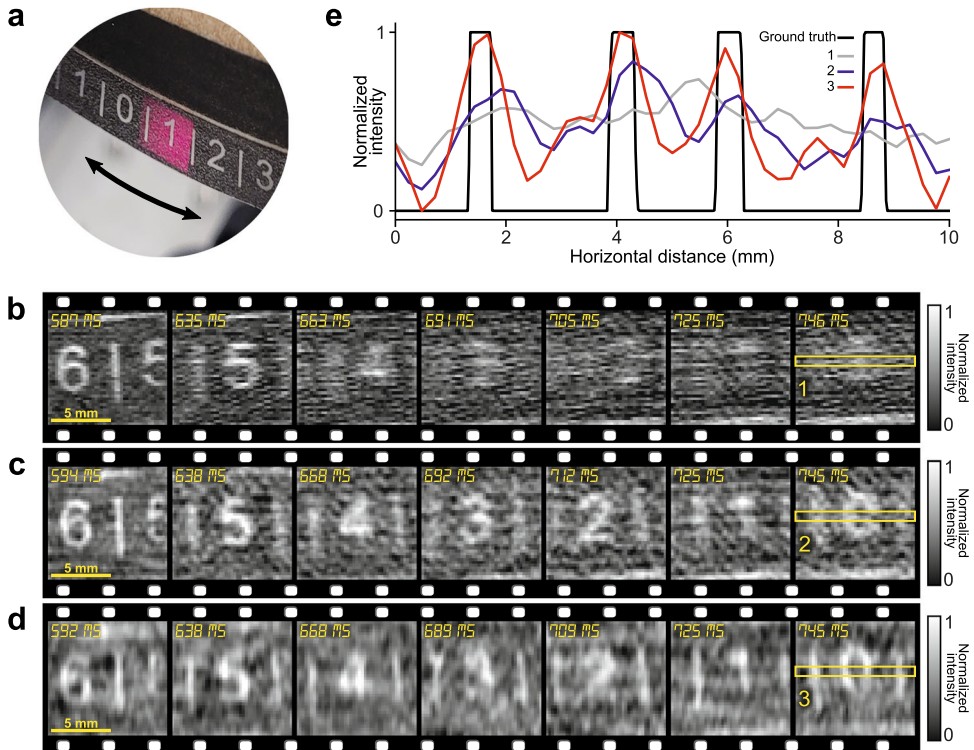

**Fig. 3 | SPI-ASAP of motion of a transmissive position ruler attached to a pendulum. a** Photograph of position ruler and illuminated FOV of 10 mm × 10 mm. **b–d** Selected frames from reconstructions with 100% sampling at 145 fps (**b**), 50%-sampling at 298 fps (**c**), and 25% sampling at 598 fps (**d**). **e** Normalized average intensity profiles from a selected region with a size of 5×43 pixels (marked by the yellow boxes in (**b–d**)), compared to the ground truth.

segmentation for video reconstruction, determination of imaging speeds, and implementation of image reconstruction using matrix multiplication are provided in Methods. Simulated comparisons of SPI-ASAP to existing CS-based reconstruction algorithms[42] are provided in Supplementary Note 4. The results of this comparison (see Supplementary Fig. 4) show that SPI-ASAP accelerates reconstruction speeds by 2–3 orders of magnitude while maintaining comparable or superior reconstructed image quality.

### Demonstration of SPI-ASAP in transmission mode

To prove the concept of SPI-ASAP, we imaged the dampened oscillations of a pendulum object with a 10 cm radius. As shown in Fig. 3a, on the bottom of this pendulum is a scale consisting of the digits "0" to "9" separated by vertical lines. "0" is placed at the pendulum's rest position, and the remaining digits are placed symmetrically at increasing distances. Working in transmission mode, the SPI-ASAP system had a frame size of 41×43 pixels ($n = 1,763$), corresponding to a field-of-view (FOV) of approximately 10 mm × 10 mm.

The motion of the pendulum was recorded over 1.54 seconds during which the pendulum exhibited dampened oscillations and came to rest at equilibrium. Supplementary Movie 1 shows three videos reconstructed from this dataset corresponding to sampling rates of 100%, 50%, and 25%, with imaging speeds of 145 fps, 298 fps, and 598 fps, respectively. Fig 3b–d show representative frames from each reconstructed video. Unlike conventional imaging for which fast-moving objects may exhibit only localized motion blurring, the sequential and wide-field nature of SPI's data acquisition produces global artifacts for highly dynamic scenes. As illustrated by the profiles compared in Fig. 3e, increased frame rates by CS-assisted SPI-ASAP reduce these blur artifacts, allowing details of fast-moving objects to be resolved. Attractively, this flexibility is controlled entirely at the reconstruction stage, thus allowing a balance between overall image quality and frame rate to be optimized to suit specific datasets.

### Demonstration of SPI-ASAP in reflection mode

To test the SPI-ASAP system in reflection mode, we imaged the internal components of a running mechanical watch movement (Fig. 4a). The balance wheel, which controlled the intermittent motion of the escape wheel, underwent sustained oscillatory motion at approximately 2.5 Hz. The SPI-ASAP system had a FOV of 14 mm × 14 mm, a frame size of 59×61 pixels, and an imaging speed of 103 fps (with 100% sampling). Fig 4b illustrates selected frames from the reconstructed videos, with the full sequence shown in Supplementary Movie 2. Intensity data from three pixels are plotted in Fig. 4c to study the motion of the internal wheels. Pixel 1 selects a position that is darkened once per full oscillation of the balance wheel and thus allows for quantification of the 2.5 Hz oscillation frequency. Pixels 2 and 3 select positions that are alternatively darkened by the passage of one tooth of the escape wheel that intermittently moves twice per balance wheel oscillation.

Another reflection-mode experiment captured the heating-induced rupture of a popcorn kernel. As shown in Fig. 4d, the kernel was held upright by a metal holder and heated by a forced stream of hot air from a heat gun. The SPI-ASAP system imaged this event at 298 fps (with 50% sampling) and with a FOV of 14 mm × 14 mm (41×43 pixels). The reconstructed movie is shown in Supplementary Movie 3 with selected frames shown in Fig. 4e. As visualized in Fig. 4f, the rupture of the kernel's shell initiated on its upper right side, with the subsequent expansion of the kernel's interior causing the kernel to be pulled from the holder by the air stream.

### SPI-ASAP in strong ambient light

Since an imaging relationship need not exist between object and detector in SPI, SPI systems can tolerate optical disruption of the imaging beam that may occur between the pattern-illuminated object and detection with the non-imaging sensor[6,25,43–45]. This characteristic well-positions SPI for scenarios requiring extreme optical filtering,

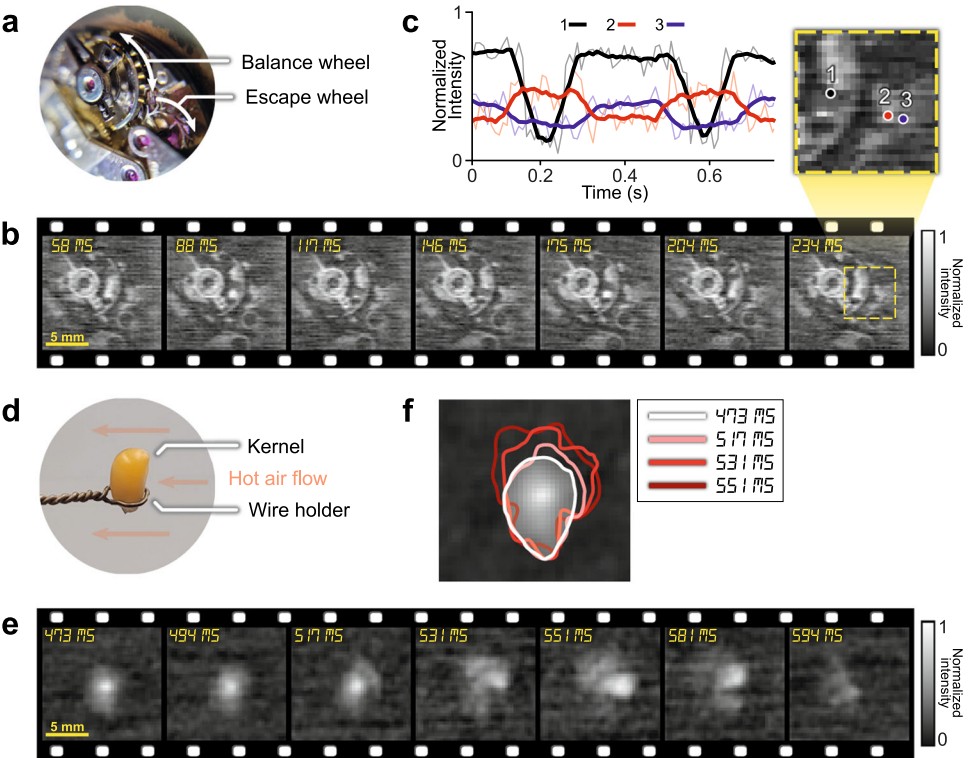

**Fig. 4 | SPI-ASAP in reflection mode. a** Photo of the internal components of the mechanical watch. **b** Selected frames of the running mechanical watch movement imaged at 103 fps. **c** Time evolution of the intensity of the three selected pixels (marked in the expanded inset from (**b**)), illustrating the approximate 2.5 Hz motion of the balance wheel and the 5.0 Hz intermittent motion of the escape wheel. Thick lines show moving averages (7-frame window) with raw data shown by the thin lines. **d** Experimental setup for imaging the rupture of a popcorn kernel. **e** Selected frames of the heating-induced rupture imaged at 298 fps with 50% sampling. **f** Traced outlines of the kernel from selected frames showing its expansion during rupture.

such as for scenes involving intense and varying ambient light. To demonstrate this capability in high-speed SPI-ASAP, we studied the current-induced failure of incandescent light bulb filaments in two experimental conditions (Fig. 5). To remove the strong and time-varying incandescence, we built spatial and chromatic filtering stages on the detection side (see details in Supplementary Note 5 and Supplementary Fig. 5). SPI-ASAP operated at 298 fps (50% sampling) and with a FOV of 11 mm × 11 mm (41×43 pixels).

The first experiment used a sealed un-modified bulb (Fig. 5a). Selected frames from the reconstructed video (Supplementary Movie 4) are shown in Fig. 5b. The burn-out of the filament occurred after 3–5 seconds of increasing incandescence. The result reveals that the failure occurred at the left connection between the filament and the supporting wire lead, followed by the deposition of the vaporized metal from the supporting wires on the interior of the glass bulb surface. This process led to a reduced transmission through the bulb, which is observed in Fig. 5b as an overall darkening following the initiation of the failure (Fig. 5c).

As a comparison, in the second experiment the cover glass of the bulb was removed (Fig. 5d). The evolution of this dynamic event is shown in Supplementary Movie 5, with selected frames presented in Fig. 5e. In contrast to the first experiment, the filament exposed to air had a quicker failure, which initiated after 1–2 seconds of increasing incandescence. Moreover, failure occurred in the filament material itself from combustion due to the presence of oxygen in air. The result also reveals the emission of smoke and vaporized material as well as the ejection of a section of filament material that fell onto and fused with the portion of glass bridging between the filament support wires. Finally, Fig. 5f shows the time evolution of the background intensity of an identical region of background pixels as that shown in Fig. 5c.

## Ultrahigh-speed SPI-ASAP at 12 kfps

For the above-discussed experiments, the operation of SPI-ASAP requires the scan-synchronized deployment of multiple aggregate patterns by the DMD, whose maximum refresh rate limits the system's performance. To further increase the imaging speed, data acquisition was carried out at the maximum scan rate of the polygonal mirror (i.e., 12 kHz) with a single static mask that combined a sufficient number of aggregate patterns to allow for reconstruction at 55% sampling (more details about the pattern generation and signal timing are included in Supplementary Note 6 and Supplementary Fig. 6). Using this experimental configuration, we imaged a 30 slot optical chopper rotating at 4800 RPM (Fig. 6a). The SPI-ASAP system imaged five different locations, each with a FOV of 10 mm × 10 mm and a frame size of 11×13 pixels.

Fig 6b–f show consecutive frames from dynamic videos (see Supplementary Movie 6) captured at 12 kfps, thus demonstrating the maximum possible frame rate of SPI-ASAP based on its current hardware. A time history of normalized intensity from a selected pixel is shown in Fig. 6g. By using a sinusoidal fitting, the wheel chop rate was determined to be 2408.9 Hz corresponding to a linear speed at the edge of the wheel of 25.7 m s$^{-1}$, in agreement with the experimental conditions.

## Demonstration of real-time SPI-ASAP

The reconstruction algorithm adopted by SPI-ASAP is well-suited for real-time operation, in which recovery and display of images must follow acquisition immediately with minimal delay. A filmed demonstration of SPI-ASAP operating in real-time for a variety of scenes, frame sizes, and frame rates is included in Supplementary Movie 7. It shows the system operating at a frame size of 59×61 pixels with a real-time display at 51 fps and 100 fps (corresponding to 100% and 50%

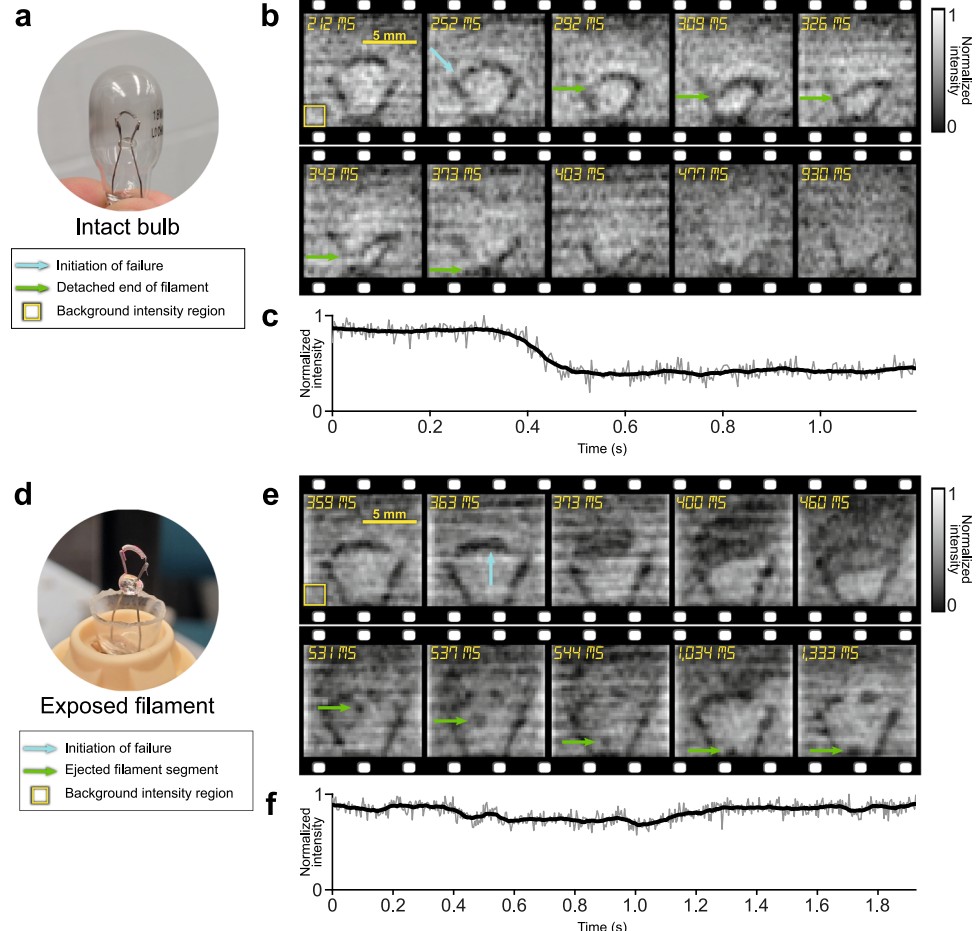

**Fig. 5 | SPI-ASAP of the burn-out of incandescent filaments. a** Photo of an unmodified bulb showing the enclosed filament. **b** Selected frames of the reconstructed video of the unmodified bulb. **c** Time evolution of the intensity of a background region (8 × 8 pixels) highlighted in (**b**). Thick line shows the moving average (21 frame window) with raw data shown by the thin line. **d** Photo of a modified bulb showing direct exposure of filament to air. **e** Selected frames of the reconstructed video of the modified bulb. **f** As (**c**), but from the background region in (**e**).

sampling, respectively), a 71×73 pixel frame size at 84 fps (50% sampling), and a 101×103 pixel frame size at 97 fps (30% sampling). GPU-computed matrix multiplications for image reconstruction typically required 0.4 ms. Data processing times, encompassing the registration of photodiode signals, image reconstruction, and display, were typically 1.0 ms.

## Discussion

We have developed SPI-ASAP based on the concept of swept aggregate patterns to enhance the capabilities of SPI for ultrahigh-speed and real-time imaging. In contrast to existing SPI methods, SPI-ASAP synergizes DMD modulation and polygonal mirror scanning to retain the practical advantages of DMDs in flexible and reconfigurable pattern projection, while extending the rate of pattern deployment by over two orders of magnitude. We have demonstrated SPI-ASAP for high-speed and real-time imaging by recording various dynamic scenes in both reflection and transmission across various frame sizes. In both the reflection and transmission modes, SPI-ASAP shows its flexibility for frame size and frame rate, resilience to strong ambient light, and its capability of ultrahigh-speed imaging at up to 12 kfps.

While currently limited by the specifications of core components, SPI-ASAP's performance could be further enhanced by optimized selections of DMD and laser scanning hardware. The pattern deployment rate of the current SPI-ASAP system is limited by the maximum display rate of the DMD (i.e., 6.37 kHz), which was approximately 53% of the 12 kHz maximum scan rate of the polygonal mirror. As discussed

in Supplementary Note 3, the scan duty cycle of the current SPI-ASAP system is compatible with an approximately 10× increase in mirror facet count, which would correspond to a maximum scan rate of 120 kHz using 45,000 RPM motorization. If the fastest DMD (with a 32 kHz refresh rate) were used, the performance of SPI-ASAP could be increased by five times, while ultrahigh-speed SPI-ASAP performance would be increased by a factor of 10. Finally, the currently deployed interpolation method for SPI-ASAP is designed to maximally leverage the parallel computing ability of GPUs. Undoubtedly, other sophisticated interpolation methods[46,47] could also be explored to potentially improve reconstruction quality while maintaining fast reconstruction.

As primarily a source of structured illumination, DMDs also impose restrictions for the optical sources compatible with SPI-ASAP, which must be compatible with both the ~10 μm size of individual DMD micromirrors and the transmission properties of the DMD's cover window. Frame size and FOV are also influenced by the DMD's size and pixel count, as well as the noise and bandwidth properties of the photodiode. Owing to the correlative behavior of the swept masks, information of image structure is represented in small variations in photodetector signals, which in general fluctuate more rapidly and with decreased amplitude as the frame size of deployed patterns is increased. Consequently, the photodiode's signal-to-noise ratio would ultimately limit frame size and scanning rate under the coding strategy of SPI-ASAP. Meanwhile, the focal lengths and clear apertures of the optics that image the DMD-displayed patterns to the object (i.e., L4–L6 in Fig. 1 and Supplementary Fig. 1) constrain the optical bandwidth of

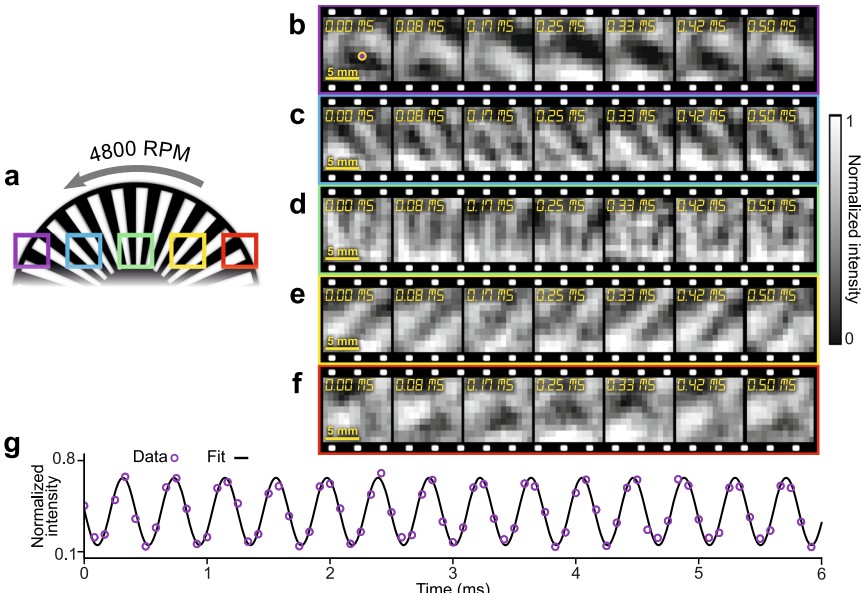

**Fig. 6 | Ultrahigh-speed SPI-ASAP of an optical chopper rotating at 4,800 RPM.**
**a** Schematic of the experimental setup. Color boxes show regions imaged by SPI-ASAP. **b–f** Consecutive frames of reconstructed ultrahigh-speed videos with outline colors corresponding to the FOV positions shown in (**a**). **g** Time history of normalized intensity for a selected pixel (marked by the purple point in (**b**)).

pattern projection, which thus limits SPI-ASAP's frame sizes by constraining the minimum pixel size of encoding patterns to be compatible with the system's optical point spread function. Additionally, timing inaccuracies caused by polygonal mirror defects as well as the continuous motion of the encoding patterns contribute to an anisotropic spatial resolution. Investigation of this property of SPI-ASAP and an analysis of contributing factors are provided in Supplementary Note 7 and Supplementary Fig. 7.

As a coded-aperture imaging modality[48], SPI-ASAP holds promises in terms of broad application scope. Besides the active-illumination-based applications demonstrated in this work, the principle of SPI-ASAP is readily applicable to high-speed imaging with passive detection, which will open new routes to applications that rely on self-luminescent events, such as optical property characterization of nanomaterials[49] and the monitoring of neural activities[50]. SPI-ASAP for imaging in strong ambient light conditions could also be adopted for the study of combustion phenomena[51]. The non-imaging relationship between object and detector in SPI-ASAP indicates its potential applications for non-line-of-sight imaging[25]. Combining SPI-ASAP with 3D profilometry will lead to the immediate application of scene relighting on high-speed 3D objects[52]. Extending the operational spectral range of SPI-ASAP will enable fast hazardous gas detection[26]. Finally, SPI-ASAP may find applications in inducing patterned regions of high conductivity in semiconductor materials for high-speed 2D THz imaging[7,10,53].

## Methods
### Geometry of beam scanning
The design of SPI-ASAP uses a rotating polygonal mirror in a double-reflection arrangement, which translates scanned illumination of the DMD surface into the swept deployment of encoding patterns. Closeups of the polygonal mirror and DMD reflections producing the desired scan behavior are illustrated in Supplementary Fig. 2. The basis of this behavior is the angular correlation of the scanned and de-scanned beams during mirror rotation. This requirement can be expressed by $\varphi_1 = \varphi_2$ for scan angles $\varphi_1$ and $\varphi_2$ (see Supplementary Fig. 2a) for light that respectively illuminates and is reflected by the DMD surface. At the position of the DMD shown in Supplementary Fig. 2b, this requirement transfers to the correlation of lateral beam

shifts as the DMD surface is scanned by illumination with an incident angle of $\theta = 24°$. If scan optics with a focal length $f_1$ are used to collimate the illuminating beam, the generated lateral shift is expressed by $d_1 = f_1 \varphi_1$. Subsequent reflections of the beam by planar mirrors (M1 and M2 in Supplementary Fig. 1a) do not alter such lateral shifts. As illustrated by the geometry of Supplementary Fig. 2b, the diffraction of the DMD produces a lateral shift of $d_2 = d_1 \sec\theta$. De-scanning optics of focal length $f_2$, re-focusing the reflected beam, then produce an angular variation of $\varphi_2 = d_2/f_2$. Combining this information produces the design requirement $f_1 = f_2 \cos\theta$, which dictated the selection of lenses L2–L4 in the SPI-ASAP system. For de-scanning, a single lens (L4) was chosen. For scanning, two lenses (L2 and L3) with specific positioning were used to produce the required effective focal length. Complete details of lens selection and positioning are included in Supplementary Note 1.

### Cyclic S-matrices
SPI-ASAP encodes measurements by using cyclic S-matrices. In general, S-matrices are defined as the class of {0,1}-valued matrices that possess a maximal possible value of the determinant[54,55]. It can be shown that for a non-trivial S-matrix $S$ of order[52,56],

$$S^{-1} = \frac{2}{n+1}\left(2S^T - J\right), \qquad (2)$$

where $J$ denotes the all-ones matrix with size $n \times n$, $S^T$ denotes the matrix transpose of $S$, and $n$ must be of the form $4k - 1$ for some positive integer $k$. In the case of cyclic S-matrices, an additional cyclic structure is imposed by which the initial row determines all subsequent rows via left-wise circular shifts. Let $S_{i,j}$ denote the element of $S$ with row index $i$ and column index $j$. The initial row $S_{0,j} (j = 0,...,n-1)$ then determines all elements of $S$ according to

$$S_{i,j} = S_{0,i+j}, \qquad (3)$$

where $i = 0, . . . ,n-1$, and the indices are interpreted modulo $n$. The consequences of the circular structure are that $S$ is symmetric (i.e. $S = S^T$) and that $S^{-1}$ is also circular. Several methods are known for the construction of cyclic S-matrices. For imaging, it is desirable that $n$ can

be factorized into parts of approximately equal size. Fortunately, given any pair of twin primes $p$ and $q = p + 2$, it is possible to construct a cyclic S-matrix of order $n = pq$[57,58].

The computation of $S_{0,j}$ depends on the following property. An integer $x$ is called a quadratic residue $(\bmod\, y)$ if $x \neq 0\,(\bmod\, y)$ and there exists another integer $z$ such that $x \equiv z^2\,(\bmod\, y)$. In this case, the following functions can be defined:

$$f(j) = \begin{cases} +1 & \text{if } j \text{ is a quadratic residue} \,(\bmod\, p) \\ 0 & \text{if } j \equiv 0\,(\bmod\, p) \\ -1 & \text{otherwise} \end{cases} \quad \text{and} \quad (4)$$

$$g(j) = \begin{cases} +1 & \text{if } j \text{ is a quadratic residue} \,(\bmod\, q) \\ 0 & \text{if } j \equiv 0\,(\bmod\, q) \\ -1 & \text{otherwise} \end{cases}, \quad (5)$$

from which $S_{0,j}$ of a cyclic S-matrix of order $n = pq$ can be computed by

$$S_{0,j} = \begin{cases} 0 & \text{if } [f(j) - g(j)]g(j) = 0 \\ +1 & \text{otherwise} \end{cases}. \quad (6)$$

## Aggregate pattern sequencing and data segmentation for video reconstruction

For video reconstruction, the smallest units of data considered for frame segmentation corresponded to the scans of individual aggregate patterns. Each scan produces a sequence of $q$ bucket signals comprising a completed row of $Y$ (Fig. 2c). Each frame of reconstructed video is then produced from a fixed number of scans (denoted by $L$) allocated into consecutive non-overlapping segments from a dataset recorded continuously during experiments. Additional scans of three registration patterns appended to the start of the pre-stored sequence (see Supplementary Note 2) are excluded during frame segmentation. As a result, the average framerate $f_r$ of recovered videos is selectable at the time of reconstruction according to the equation

$$f_{!r} = \begin{cases} \frac{f_{!s}}{L+3} & \text{if } L = p \\ \frac{f_s}{L+3\frac{L-1}{p}} & \text{if } L < p \end{cases}, \quad (7)$$

where $f_{!s}$ is the scan rate of the polygonal mirror which for our experiments (except for ultrahigh-speed imaging), was 6.37 kHz, limited by the maximum refresh rate of our DMD.

In operation, a complete sequence of aggregate patterns is pre-stored by the DMD and displayed iteratively. To support under-sampling, aggregate patterns are displayed in a permuted order that allows groups of $L < p$ consecutive scans to fill the rows of $Y$ with approximately uniform spacing. By numbering the aggregate patterns with $k = 0, \dots, p - 1$ corresponding to the rows of $Y$, the display of the aggregate patterns took place in the order $d_0, d_1, \dots d_{p-1}$ where the sequence of values $d_k$ was defined by

$$\begin{aligned} d_0 &= 0, \text{ and} \\ d_{k+1} &= d_k + c \quad (\bmod\, p). \end{aligned} \quad (8)$$

Here, $c$ is a constant chosen from an inspection of the sampling uniformity for various values of $L$. As a consequence of using cyclic S-matrices derived from the twin-prime construction, $p$ is prime, and thus Eq. (8) defines a permutation for any value of $c \neq 0\,(\bmod\, p)$. The preferred values of $c$ used for our experiments are summarized in Supplementary Table 2.

## Interpolation-based image reconstruction using matrix operations

To take full advantage of GPU hardware for real-time visualization, SPI-ASAP's image reconstruction is developed based on matrix multiplication for direct parallelization. In general, it consists of two computational steps: interpolation and image recovery. For a segment of $L$ consecutive scans deployed in a permuted order, recorded data are represented by a $L \times q$ matrix $M$ whose elements $M_{ij}$ respectively denote the $j^{\text{th}}$ bucket signal ($j = 0, \dots, q - 1$) acquired from the deployment of the $i^{\text{th}}$ aggregate pattern ($i = 0, \dots, L - 1$).

The goal of interpolation is to transform $M$ into an estimate of the smooth bucket signal matrix $Y$. This transformation can be computed by matrix product

$$Y = WMV, \quad (9)$$

where the matrices $V$ (size $q \times q$) and $W$ (size $p \times L$) are carefully designed to carry out optional row-wise low-pass filtering, and column-wise interpolation, respectively. Both the matrices $V$ and $W$ can be pre-computed and stored with low overhead owing to their sizes scaling linearly to the frame size of reconstructed images. Full details of the pre-computations involved for the elements of the matrices $W$ and $V$ are provided in Supplementary Note 8 and Supplementary Fig. 8.

Then, by reshaping $Y$ back into an $n$-element vector $\mathbf{y}$, the reconstruction of a 2D image $\mathbf{x}$ then follows by direct inversion according to Eq. (1). Because the structure of $S^{-1}$ is cyclic, computation of $\mathbf{x}$ becomes equivalent to a convolution of $\mathbf{y}$ with the initial row of $S^{-1}$,

$$\mathbf{x} = \mathbf{y} * \mathbf{s}_0^{-1}, \quad (10)$$

where $\mathbf{s}_0^{-1}$ is the initial row of $S^{-1}$, and the operator $*$ denotes discrete circular convolution.

During real-time operation, parallelization of reconstruction computations was facilitated with the use of a GPU (GeForce GTX 1060, NVIDIA). Frame rates were limited by the rate of data acquisition from scanning, with reconstruction and display taking place quickly enough to consume all incoming data.

## Data availability

All data needed to evaluate the findings of this study are present in the paper and Supplementary Information. Raw data for Fig. 3 and 6 are provided with the software package described in the code availability statement. All other raw data in this study are available from the corresponding author upon request.

## Code availability

A MATLAB software implementation of SPI-ASAP is available at https://doi.org/10.5281/zenodo.7227472.

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

## Acknowledgements

The authors thank Jeremy Gribben and Alan Boate from Ajile Light Industries and Cheng Jiang from Institut National de la Recherche Scientifique for technical assistance. J.L. acknowledges the support by the Natural Sciences and Engineering Research Council of Canada (NSERC) (RGPIN-2017-05959, RGPAS-2017-507845, I2IPJ-555593-20), Canada Foundation for Innovation and Ministère de l'Économie et de l'Innovation du Québec (37146), Canadian Cancer Society (707056), New Frontier in Research Fund (NFRFE-2020-00267), Fonds de Recherche du Québec–Nature et Technologies (203345 - Centre d'optique, photonique et lasers), and Fonds de Recherche du Québec–Santé (267406, 280229). P.K. acknowledges the support from the Alexander Graham Bell CGS-D scholarship from NSERC.

## Author contributions

P.K., J.L., and T.O. conceived the idea. P.K. created the experimental setup and developed all software. P.K. performed the experiments and data analysis. P.K. wrote the manuscript, with J.L. and T.O. providing editorial input. J.L. proposed the concept, contributed to experimental design, and supervised the project.

## Competing interests

The authors disclose the patent application of US Provisional 63/380,620 (P.K., J.L., and T.O.).
