## [Peer Review File · Nature Communications]

Compressed ultrahigh-speed single-pixel imaging by swept aggregate patternsREVIEWER COMMENTS

Reviewer #1 (Remarks to the Author):

This work developed a novel modulation acceleration technique termed swept aggregate patterns to reach pattern projection rates of up to 14.1 MHz. The authors have achieved real-time single-pixel imaging (SPI) at 96 fps with a resolution of 101×103 pixels using the method described. It bypasses the speed restriction of DMDs in traditional SPI setups and will serve as a crucial building block for future real-time or large-scale SPI equipment. This paper is quite original and has significant scientific worth. I have comments as follows:

- a) A fundamental formula considering the relationship between modulation speed, image resolution, and sampling ratio (even noise level if possible) should be provided for the further development of SPI. According to the aforementioned formula and the system settings in this work, one can determine information such as the maximum imaging speed under the acceptable resolution, or maximum imaging resolution meeting the minimum requirements for real-time imaging at 24fps.
- b) What are the factors that limit the modulation speed and resolution for the proposed swept aggregate patterns? What potential future innovations are available to further improve the reported technique?
- c) It is better to compare the computing efficiency and accuracy of the reported reconstruction to other CS-based SPI algorithms (10.1364/JOSAA.35.000078).

Reviewer #2 (Remarks to the Author):

In this paper, the authors proposed a method of single-pixel imaging for high-speed monitoring of fast phenomena. Although many techniques of single-pixel imaging have been reported resultant of recent developments in optical and computational technologies, tough restrictions in the data acquisition with a single pixel detector remain as a growth bottleneck in achieving real-time/high-speed monitoring, which is required in many applications. The authors have achieved a breakthrough in single-pixel imaging by providing new techniques in both hardware and software of the imaging system in a practical manner. The authors performed the proof-of-principle experiment by performing real-time/high-speed single-pixel imaging of dynamic events with various conditions. The optical system consists of the fundamental optics: lens (which can be replaced by concave mirrors) and mirrors. It means that the proposed method can be applied not only in visible range but also in non-visible ranges, including sound and other physical probes where optical instruments such as an image sensor and spatial light modulator do not exist.

I think this work is interesting and significant enough for publication in Nature Communications. The manuscript is clearly written. The characterization of the proposed method and experimental demonstrations are sufficient to show its capability. Therefore, I recommend publication in Nature Communications, provided that the authors address the following comments and concerns:

1. In the proposed method, is the spatial resolution (not the pixel resolution) of the reconstructed images the same in the vertical and horizontal directions?
2. In the first demonstration experiment (Fig. 3), how the best sampling is determined? As the authors mentioned, the higher frame rate reduces the effect of motion blur, but lower sampling decreases image quality. There is a trade-off.
3. Can frame rates be optimized at any time depending on the scene? The frame rate may

no longer need to be constant within the same video?

4. I recommend the authors highlight the advantages of the proposed method over a 2D image sensor based on the features and performance achieved (or will be achieved) in this work. Especially in the Discussion, the authors claim the improvement in the performance of the proposed imaging method by using high-end DMDs. It may require higher costs and a larger system, and some readers may think the proposed method loses the advantages compared with imaging with a 2D image sensor.

5. Could you support your claim “SPI-ASAP holds promises in terms of broad application scope.” in the Discussion by adding more information about the features of the proposed method and/or information about possible applications?

RESPONSES TO REVIEWERS

We sincerely appreciate both reviewers for their careful and thorough evaluation of our manuscript. We have carefully adopted the reviewers' comments to improve the quality of our manuscript. In this round of revision, we have included detailed descriptions to explain and compare the technical features of our work. We have also conducted many new experiments to directly address the reviewers' concerns. These amendments are reflected in the new information and clarification presented in Main Text. As for Supplementary Materials, the major changes include the following points:

- We have added a new **Supplementary Note 4** and **Supplementary Fig. 4** to demonstrate the superior performance of the image reconstruction method used in SPI-ASAP compared to two widely used algorithms based on compressed sensing.
- We have added a new **Supplementary Note 7** and **Supplementary Fig. 7** to thoroughly evaluate the spatial resolution of the SPI-ASAP system.

The introduction of these new supplementary notes has entailed a re-labeling of some supplementary note sections in the revised manuscript. The point-by-point responses are listed as follows. All changes are highlighted in red text in the revised manuscript.

REVIEWER #1

[Comment 0]

This work developed a novel modulation acceleration technique termed swept aggregate patterns to reach pattern projection rates of up to 14.1 MHz. The authors have achieved real-time single-pixel imaging (SPI) at 96 fps with a resolution of 101×103 pixels using the method described. It bypasses the speed restriction of DMDs in traditional SPI setups and will serve as a crucial building block for future real-time or large-scale SPI equipment. This paper is quite original and has significant scientific worth.

[Response 0]

We deeply thank the reviewer for acknowledging the novelty and significance of our work, and for sharing our perspective of SPI-ASAP as a building block for the future development of SPI.

[Comment 1]

A fundamental formula considering the relationship between modulation speed, image resolution, and sampling ratio (even noise level if possible) should be provided for the further development of SPI. According to the aforementioned formula and the system settings in this work, one can determine information such as the maximum imaging speed under the acceptable resolution, or maximum imaging resolution meeting the minimum requirements for real-time imaging at 24fps.

[Response 1]

We believe that our manuscript has already supplied such a formula in equation 7 in Methods (see Main Text, Lines 365–368). We repeat this formula below for convenience:

$$f_r = \begin{cases} \frac{f_s}{L+3} & \text{if } L = p \\ \frac{f_s}{L+3\frac{L-1}{p}} & \text{if } L < p \end{cases} \quad (7)$$

This formula establishes the average framerate f_r of recovered video for high-speed and/or real-time SPI-ASAP, as a function of vertical frame size p , number of recorded scans L , and the polygonal mirror scan rate f_s that was fixed at 6.37 kHz. The sampling ratio of such an arrangement is therefore L/p , and the horizontal frame size q via the twin-prime construction of our encoding patterns (see Main Text, Lines 350–352) satisfies $q = p + 2$. Finally, we note that as described in Main Text (Lines 363–365), the exclusion of the three registration patterns from frame segmentation resulted in a slightly non-uniform frame period. This effect is however captured by the above formula by reporting an average.

The demonstration of ultrahigh-speed SPI-ASAP, on the other hand, entailed a simplified relationship between scan rate and frame rate, for which $f_r = f_s$, with f_s held at the maximum scan rate of our polygonal mirror of 12 kHz (see Main Text, Lines 236–243).

In addition to these formulas, we note that additional constraints on the imaging frame size $p \times q$ are imposed by the finite pixel count of the DMD. In particular, the DMD must be capable of displaying the individual aggregate patterns of $p \times (2q - 1)$ in size (see Main Text, Lines 132–134). For the ultrahigh-speed configuration, fitting multiple aggregate patterns within one DMD pattern results in a more constrained range of acceptable frame sizes, with the dimensions of the total pattern scaling in the way depicted in Supplementary Fig. 6.

Therefore, the formulas and information described in the manuscript can help readers determine the theoretical maximum imaging speed under various system settings. They also allow

calculating the maximum frame size compatible with real-time imaging at 24 fps, which is 239×241 pixels for full sampling using cyclic S-matrices.

[Comment 2]

What are the factors that limit the modulation speed and resolution for the proposed swept aggregate patterns? What potential future innovations are available to further improve the reported technique?

[Response 2]

The maximum modulation speed of SPI-ASAP is determined jointly by hardware limitations, signal processing factors, and horizontal frame size. From the aspect of hardware, the two most important limitations arise from the DMD and the polygonal mirror. The analysis can be separated for the two working modes of SPI-ASAP. First, the high-speed and/or real-time operation of SPI-ASAP depends on the synchronized operation of the DMD and polygonal mirror. The smaller value of the maximum DMD refresh rate and the maximum polygonal mirror scan rate determines the scan rate of SPI-ASAP. As we have described in Main Text (Lines 367–368), the DMD refresh rate (i.e., 6.37 kHz) determined this limit in this working mode. Second, in the ultrahigh-speed operation of SPI-ASAP (see Fig. 6, Main Text Lines 239–243, and Supplementary Fig. 6), the DMD remains static during the operation. Therefore, the maximum modulation speed solely depends on the polygonal mirror (i.e., 12 kHz).

In addition to the above limitations in hardware, we also mention the following factors in signal processing. Although not limiting the current system, they must be considered to accommodate modulation speed for actual imaging:

- Photodetector bandwidth: this bandwidth should satisfy a temporal response consistent with a sampling period below that of the time taken for the optical sweeping to shift the aggregate patterns by one encoding pixel. For the photodetector used in the SPI-ASAP system (see the description in Supplementary Materials, Lines 36–39), the use of a transimpedance amplifier meant that the limit of the measurement bandwidth was determined by the specifications of the photodiode (25 MHz). As we have reported in Supplementary Table 1, this bandwidth was sufficient for the peak modulation rates (i.e., 14.1 MHz) of SPI-ASAP.

- Digital sampling rate: Our current digitization equipment supports digital sampling rates of up to 500 MHz. This rate is sufficient for the required sampling rate of 20 MHz for high-speed and/or real-time operation (see Supplementary Materials, Lines 47–51) and 50 MHz for ultrahigh-speed operation (see Supplementary Materials, Lines 175–176).

Finally, we note that the modulation rate for SPI-ASAP is ultimately determined by both the scan rate (either 6.37 kHz or 12 kHz for the current system) and the horizontal frame size q . For high-speed and/or real-time SPI-ASAP, each scan deploys q encoding patterns (see Main Text, Lines 129–132). For ultrahigh-speed imaging, this number is Kq , where K is the number of aggregate patterns composed simultaneously in the multi-aggregate pattern (see Supplementary Materials, Lines 167–169, and Supplementary Fig. 6).

Regarding the maximum frame sizes feasible for SPI-ASAP, we would like to point out that they are also influenced by modulation rates, bandwidth limitations, and scanning motion. First, as we have noted above, the modulation rate of SPI-ASAP is proportional to the horizontal image frame size, and thus frame size limitations are interrelated with the sampling limitations affecting modulation speed. In other words, the higher modulation speeds that necessarily accompany larger frame sizes must be compatible with the hardware used for photodetection and digitization. Second, the frame size should be compatible with the DMD’s pixel count for the display of appropriately sized aggregate patterns (see also our response to this reviewer’s Comment 1). Third, we itemize the following bandwidth limitations that impact frame size:

- Signal-to-noise ratio (SNR): As we have noted in Main Text (Lines 291–294), the sampling behavior of the swept masks, as a form of image convolution (Main Text, Lines 137–140), means that increased frame sizes will produce photodetector signals exhibiting fluctuations of increased frequency and decreased amplitude. Thus, photodiode SNR must ultimately limit frame size and scanning rate under the coding strategy of SPI-ASAP.
- Optical bandwidth: The focal lengths and clear apertures of the optics that image the binary patterns displayed on the DMD surface to the object (i.e. L4–L6 in Fig. 1, and Supplementary Fig. 1) ultimately constrain the optical bandwidth of pattern projection, and so must also limit SPI-ASAP’s frame sizes by constraining the minimum size of encoding pixels to be compatible with the system’s optical point spread function.

Finally, we note that other sources (e.g., the imperfections of the polygonal mirror and the continuous motion of the encoding patterns) affect the measured spatial resolution of SPI-ASAP. Please see details in our response to Comment 1 of Reviewer 2.

As for the last part of the reviewer's comment, we list the following future innovations as being amenable to the improvement of SPI-ASAP:

- Higher performing DMD hardware: As discussed above and in Main Text (Lines 276–278), a DMD with a faster refresh rate and a larger micromirror array size would encompass improvements to both resolution and display rates.
- Higher performing polygonal mirror: As discussed above and in Main Text (Lines 278–281), a polygonal mirror with more facets would encompass both maximum motorization speed and facet count.
- More sophisticated interpolation methods: Although the interpolation method that we designed for SPI-ASAP maximizes GPU performance (see Main Text, Lines 157–161), we speculate the heuristic role of interpolation in our reconstruction algorithm makes SPI-ASAP compatible with other interpolation methods (such as the ones using deep neural networks) that may improve reconstruction quality with comparable performance.

[Comment 3]

It is better to compare the computing efficiency and accuracy of the reported reconstruction to other CS-based SPI algorithms (10.1364/JOSAA.35.000078).

[Response 3]

We thank the reviewer for suggesting this work (<https://doi.org/10.1364/JOSAA.35.000078>), which presents numerical simulations comparing the performance of various reconstruction techniques for SPI. After having carefully reviewed this work in relation to SPI-ASAP, we have simulated results comparing the performance of SPI-ASAP to two widely used reconstruction methods based on compressed sensing (CS). We selected the two CS algorithms used in our comparison specifically based on the survey of SPI reconstruction methods presented in the recommended article. Specifically, we cover both kinds of regularization priors within the class of non-linear iterative methods examined in the paper.

We have incorporated these results in the present manuscript in the newly added Supplementary Note 4 and Supplementary Fig. 4. These changes have also led to the inclusion of the recommended article in the revised Main Text.

REVIEWER #2

[Comment 0]

In this paper, the authors proposed a method of single-pixel imaging for high-speed monitoring of fast phenomena. Although many techniques of single-pixel imaging have been reported resultant of recent developments in optical and computational technologies, tough restrictions in the data acquisition with a single pixel detector remain as a growth bottleneck in achieving real-time/high-speed monitoring, which is required in many applications. The authors have achieved a breakthrough in single-pixel imaging by providing new techniques in both hardware and software of the imaging system in a practical manner. The authors performed the proof-of-principle experiment by performing real-time/high-speed single-pixel imaging of dynamic events with various conditions. The optical system consists of the fundamental optics: lens (which can be replaced by concave mirrors) and mirrors. It means that the proposed method can be applied not only in visible range but also in non-visible ranges, including sound and other physical probes where optical instruments such as an image sensor and spatial light modulator do not exist.

I think this work is interesting and significant enough for publication in Nature Communications. The manuscript is clearly written. The characterization of the proposed method and experimental demonstrations are sufficient to show its capability. Therefore, I recommend publication in Nature Communications, provided that the authors address the following comments and concerns: ...

[Response 0]

We thank the reviewer for his/her thoughtful assessment of our work, in particular for sharing our insight into how the design of SPI-ASAP enhances its applicability to non-optical imaging scenarios.

[Comment 1]

In the proposed method, is the spatial resolution (not the pixel resolution) of the reconstructed images the same in the vertical and horizontal directions?

[Response 1]

We thank the reviewer for providing this discerning question. In response, we have carried out additional experiments to quantify and compare the spatial resolution of SPI-ASAP in the horizontal and vertical directions. We found that the spatial resolutions in these two directions are

different. The results of our resolution quantification experiment have now been incorporated into our manuscript in the form of an addition in Main Text (see Lines 300–303), together with a new supplementary note (Supplementary Note 7) and a new figure (Supplementary Fig. 7). We also now distinguish between “frame size” (i.e. pixel dimensions) and “resolution” in all parts of the manuscript. Indeed, we observe a lower resolution in the horizontal direction. We have identified and quantified two elements that can account for this result. Summarizing the contents of the new Supplementary Note 7, these are

- the motion of the encoding masks during the period sampled for the registration of bucket signals, and
- timing inaccuracies caused by fabrication imperfections in the facet geometry of the polygonal mirror.

In future work, the correction and/or mitigation of these factors will probably be included in SPI-ASAP reconstruction via additional steps taken in system calibration and processing.

[Comment 2]

In the first demonstration experiment (Fig. 3), how the best sampling is determined? As the authors mentioned, the higher frame rate reduces the effect of motion blur, but lower sampling decreases image quality. There is a trade-off.

[Response 2]

Respectfully, we wish to correct an underlying assumption made in this comment regarding the content of Fig. 3 in our manuscript and the methodology behind our demonstration of the variable framerate of SPI-ASAP. In particular, we stress that the results presented in Fig. 3 (see also Main Text, Lines 177–187) do not attempt to quantify a reconstruction framerate that “best” reconstructs the underlying scene (we also believe this is what the reviewer was referring to by “best sampling”). Rather, the results in Fig. 3 are intended to illustrate the decrease in motion-induced artifacts exhibited by reconstructions obtained at rounded non-optimized sampling ratios (i.e., 100%, 50%, and 25%). To enhance the presentation of Fig. 3, we extracted profiles of normalized intensity that compared the reconstruction results against the ground truth. By increasing the framerate at the reconstruction stage, we show that the resulting comparison of line intensity

profiles (Fig. 3e) demonstrates an improved ability to resolve the details of this scene under the conditions of motion blurring.

Second, the reviewer has correctly identified the trade-off between the sampling and the frame rate. Nevertheless, the prerequisite of being able to analyze this trade-off is having prior information on the targeted scene, in particular, the spatial feature size and the moving speed. The former would determine the required spatial resolution while the latter would suggest the needed modulation rate. Then, by considering the system's specifications and by using a merit function (e.g., PSNR or SSIM), the best imaging strategy could be determined. However, in most cases, we do not have the prior information on the scene, which makes the goal of such optimization generally infeasible. Instead, we only emphasize the trade-off between image quality and frame rate as an attractive feature of flexible control that SPI-ASP can afford to experimenters so that users may select parameters to determine the best combination at the time of reconstruction. In this capacity, we believe that Fig. 3 offers a sufficient range of coverage to illustrate this trade-off.

[Comment 3]

Can frame rates be optimized at any time depending on the scene? The frame rate may no longer need to be constant within the same video?

[Response 3]

For SPI-ASAP, the answer to both questions is yes. In addition to reiterating and highlighting our comments about this feature in the present manuscript (see Main Text, Lines 183–187 and 359–368), we offer some further explanations below.

For video imaging under SPI, collected data consists of a stream of bucket signal measurements from the detector recorded in tandem with the deployment of encoding patterns. In contrast to 2D sensors, which necessarily partition data encompassing a single captured “frame” according to the dimensions of the sensor array, an SPI system equipped with a suitable reconstruction algorithm may regard recorded measurements as a continuous input stream that can be partitioned arbitrarily, with each partition giving rise to a movie frame following reconstruction. Our demonstration of frame-rate modification (see Fig. 3 and Main Text, Lines 177–187) shows three separate movies derived from the same recorded data, each with an individually constant

framerate. Indeed, although our current results do not explore this possibility, a movie may be recovered from recorded SPI data which has a non-constant framerate.

We would further highlight the benefits of this flexibility for the case of real-time operation of SPI-ASAP. As described in Methods (Lines 369–378 in Main Text), we permute our sequence of DMD-displayed aggregate patterns to support the flexible frame rate in reconstruction in accordance with the interpolation step (see Main Text, Lines 151–159). This flexibility is what enables the selection of frame rate during reconstruction to be completely isolated from the DMD pattern sequence. Accordingly, our real-time software implementation (Supplementary Video 7, and Main Text, Lines 253–262) also benefits from this flexibility, supporting user-selectable on-the-fly changes to the frame rate of displayed videos without modifying the physical system.

[Comment 4]

I recommend the authors highlight the advantages of the proposed method over a 2D image sensor based on the features and performance achieved (or will be achieved) in this work. Especially in the Discussion, the authors claim the improvement in the performance of the proposed imaging method by using high-end DMDs. It may require higher costs and a larger system, and some readers may think the proposed method loses the advantages compared with imaging with a 2D image sensor.

[Response 4]

SPI-ASAP has the following features compared to a 2D image sensor. First, with a frame rate of up to 12 kfps, the SPI-ASAP system possesses a higher imaging speed compared to many 2D image sensors. Compared to representative high-speed cameras (e.g., FASTCAM SA-Z with a price of ~US \$150,000), SPI-ASAP is considerably more economical (with a total cost of <US \$10,000). Second, although not demonstrated in this work, SPI-ASAP possesses a multiplexing advantage. Because of the spatial-integration operation in data acquisition, both the raw measurement and the reconstructed image can have a better signal-to-noise ratio compared to the 2D image sensor under the same experimental settings.

We also would like to emphasize that SPI fundamentally aims to overcome the technological challenge encountered when practical constraints prohibit the use of 2D image sensors (e.g., low or no response in certain spectral ranges). This distinction is of paramount

importance when considering potential performance comparisons of SPI with that of 2D sensors. Instead of being regarded as mere alternatives in competition with conventional 2D image sensors, SPI is more suited as an independent technology primarily developed to operate beyond their scope. The coverage of SPI's technological features compared to 2D image sensors is highlighted by the following sentence in Main Text (Lines 25–27): “*By eliminating the need for a 2D (e.g., CCD or CMOS) sensor, SPI may use detectors whose cutting-edge performance or high specialization are impractical to manufacture in an array format.*”

Finally, we address the reviewer's comment regarding the costs/constraints incurred by the use of a high-end DMD. Although such hardware could be expected to increase system cost, the current design of SPI-ASAP is fully compatible with any DMD and controller hardware whose physical dimensions do not extend past the DMD chip surface (see Fig. 1 or Supplementary Fig. 1, which are scale-accurate to our current system). To our knowledge, this constraint adheres to all DMD evaluation modules that are commercially available. Moreover, high-end DMDs, which use either 0.7” or 0.95” DMD chips, are only slightly bigger than the 0.45” DMD used in the SPI-ASAP system. Considering that most space on a DMD module is occupied by electronics, upgrading the DMD will not increase the overall size of the system. Finally, a high-end DMD will cost US \$10,000–20,000. Even with the implementation of such a device, SPI-ASAP will remain cost-efficient compared to existing high-speed cameras. Thus, replacement of our current DMD would not incur any increased costs beyond those associated with the hardware itself and in particular would not entail the selection of different optical components or result in a larger system.

[Comment 5]

Could you support your claim “SPI-ASAP holds promises in terms of broad application scope.” in the Discussion by adding more information about the features of the proposed method and/or information about possible applications?

[Response 5]

We would first highlight that the current manuscript supports our claim of “broad application scope” by suggesting an important generalization of the system design (see Main Text, Lines 305–308) and proposing two concrete applications in separate scientific fields (see Main Text, Lines

308–310 and 314–315). To further support this claim, we have added more information about the features of the SPI-ASAP system and its possible applications to the revised manuscript.

With regards to the generalization to passive detection, this functionality allows the SPI-ASAP system to remove the requirement that a target receives structured illumination, and thus clearly broadens the scope of applications for SPI-ASAP by allowing it to function as a receive-only camera. This broader scope of application for passive detection-capable systems holds in particular when compared to SPI approaches based on illumination modules built with LEDs, as we have already noted in Main Text (Lines 54–56). As a coded-aperture imaging modality [R1], SPI-ASAP will provide an experimental platform to test various deep neural networks [R2]. In addition, with high coding flexibility and a high imaging speed, the passive system could be implemented in many areas of study, including particle velocimetry [R3], optical property characterization of nanomaterials [R4], and the monitoring of neural activities [R5]. An expanded discussion of applications for the passive-mode operation has been added to Lines 305–308 in the revised manuscript.

As for the first of our two proposed applications, we have suggested the use of SPI-ASAP for the study of combustion phenomena (see Main Text Lines 308–310). Based on our experimental demonstration of SPI-ASAP in strong ambient light (see Fig. 5 and Lines 209–234 in Main Text, Supplementary Fig. 5, and Lines 146–160 in Supplementary Materials), we regard this application as being particularly well-supported by the current manuscript. In particular, we would highlight the following rationale (Lines 210–214 in Main Text): *“Since an imaging relationship need not exist between object and detector in SPI, SPI systems can tolerate optical disruption of the imaging beam that may occur between the pattern-illuminated object and detection with the non-imaging sensor. This characteristic well-positions SPI for scenarios requiring extreme optical filtering, such as for scenes involving intense and varying ambient light.”* In this direction, other potential applications include hazardous gas detection [R6], non-line-of-sight imaging [R7], and scene relighting [R8]. The new information and references have been added to the revised Main Text (see Lines 310–313).

The second direction of application is high-speed 2D THz imaging (see Main Text, Lines 314–315). The main motivation of this application is to implement SPI-ASAP for high-speed imaging and sensing in a spectral range where a high-speed 2D image sensor is not available. Examples of high-speed THz imaging work leveraging SPI can be found in Refs. [7], [10], and

[53] cited in Main Text. In particular, these methods use structured illumination (with either continuous-wave or pulsed lasers) to modify the conductivity (and thus the THz absorption properties) of semiconductor substrates, which allows spatial modulation of freely propagating THz beams. We note that the basis of SPI modulation used in these works was provided by DMD-based structured illumination, directly mirroring the operation of SPI-ASAP. In envisioning such a THz application of SPI-ASAP, we anticipate that improvements to imaging speed will shed light on the development of a real-time 2D THz camera.

REFERENCES

- [R1] Liang, J. Punching holes in light: recent progress in single-shot coded-aperture optical imaging. *Rep. Prog. Phys.* **83**, 116101 (2020).
- [R2] Marquez, M. *et al.* Deep-learning supervised snapshot compressive imaging enabled by an end-to-end adaptive neural network. *IEEE J. Sel. Top. Signal Process.* **16**, 688-699 (2022).
- [R3] Liu, X. *et al.* Single-shot real-time compressed ultrahigh-speed imaging enabled by a snapshot-to-video autoencoder. *Photonics Res.* **9**, 2464-2474 (2021).
- [R4] Liu, X. *et al.* Fast wide-field upconversion luminescence lifetime thermometry enabled by single-shot compressed ultrahigh-speed imaging. *Nat. Commun.* **12**, 6401 (2021).
- [R5] Zhang, Y. *et al.* Ultrafast and hypersensitive phase imaging of propagating internodal current flows in myelinated axons and electromagnetic pulses in dielectrics. *Nat. Commun.* **13**, 5247 (2022).
- [R6] Gibson, G. M. *et al.* Real-time imaging of methane gas leaks using a single-pixel camera. *Opt. Express* **25**, 2998-3005 (2017).
- [R7] Musarra, G. *et al.* Non-line-of-sight three-dimensional imaging with a single-pixel camera. *Phys. Rev. Applied* **12**, 011002 (2019).
- [R8] Kilcullen, P., Jiang, C., Ozaki, T. & Liang, J. Camera-free three-dimensional dual photography. *Opt. Express* **28**, 29377-29389 (2020).

REVIEWERS' COMMENTS

Reviewer #1 (Remarks to the Author):

The authors have addressed the reviewer's concerns, and this manuscript is recommended for publication in Nature Communications.

Reviewer #2 (Remarks to the Author):

All of my comments and concerns are well addressed.

RESPONSES TO REVIEWERS

REVIEWER #1:

[Comment]

The authors have addressed the reviewer's concerns, and this manuscript is recommended for publication in Nature Communications.

[Response]

We thank the reviewer for acknowledging that we have addressed the concerns and for supporting the publication of this manuscript.

REVIEWER #2:

[Comment]

All of my comments and concerns are well addressed.

[Response]

We thank the reviewer for acknowledging that we have well addressed all the comments and concerns.